# Cognitive Function during and after Pregnancy and One-Year Postpartum in Type 1 Diabetes: A Longitudinal Study

**DOI:** 10.3390/nu16162751

**Published:** 2024-08-17

**Authors:** Marina Ivanisevic, Vesna Elvedji Gasparovic, Mislav Herman, Josip Delmis

**Affiliations:** Department of Obstetrics and Gynecology, University Hospital Centre Zagreb, School of Medicine, University of Zagreb, 10000 Zagreb, Croatia; marina.ivanisevic@pronatal.hr (M.I.); vesnagasparo@gmail.com (V.E.G.); mislav.herman@gmail.com (M.H.)

**Keywords:** cognitive function, type 1 diabetes mellitus, pregnancy, postpartum, leptin, body mass index, reasoning, attention

## Abstract

Background. This study aims to compare the cognitive function of women with T1DM during and after pregnancy, as well as one year post-delivery. Additionally, it aims to investigate the impacts of leptin and body mass index on cognitive function. Methods. A prospective longitudinal cohort study was conducted involving 64 pregnant women with T1DM. Cognitive function was assessed using a cognitive assessment battery during the first trimester, immediately after delivery, and one year postpartum for the final assessment. This program evaluates a wide range of cognitive abilities and provides a comprehensive cognitive well-being score (high–moderate–low), identifying strengths and weaknesses in reasoning, memory, attention, coordination, and perception. Results. The average age of the participants was 30.9 years, with a mean diabetes duration of 14.9 years. Pregnant women with a BMI of 30 kg/m^2^ or higher faced an increased risk of reduced cognitive function, memory, and reasoning. Additionally, mothers with lower overall cognitive function and memory levels had significantly higher concentrations of leptin in their blood. Conclusions. Cognitive functions—particularly reasoning and attention—are adversely affected in women with T1DM during pregnancy and shortly after delivery. Elevated BMI and leptin levels can be linked to worse cognitive outcomes in this population.

## 1. Introduction

Cognitive functions encompass a broad range of mental abilities that are essential for carrying out daily activities. These include memory, problem-solving, attention, coordination, and perception, all of which play a crucial role [1]. Cognitive impairments can significantly affect quality of life and daily functioning, making it crucial to understand these factors in an individual’s ability to function effectively in various aspects of life [1]. Cognitive impairment can profoundly impact an individual’s quality of life, affecting everything from basic daily tasks to more complex decision-making processes. T1DM results from the immune-mediated destruction of β-cells, leading to an inability to produce or secrete insulin [2]. Insulin is a hormone that plays a vital role in regulating blood glucose levels, and its absence leads to hyperglycemia, a hallmark of diabetes. Pregnancy poses unique challenges for women with T1DM, requiring careful management to ensure the health of both mother and newborn. Hyperglycemia is a key feature of T1DM due to the cessation of endogenous insulin secretion, while exogenous insulin administration can lead to hypoglycemia. Prior research has shown that individuals with type 1 diabetes often experience diminished cognitive functions, including learning, memory, attention, information processing speed, and visual perception, compared with non-diabetic individuals [3,4,5,6]. The impact of these impairments is not uniform; it varies significantly based on factors such as the age of the individual and the duration of the disease. Younger individuals and those with a longer history of diabetes tend to exhibit more pronounced cognitive deficits [4]. In an insightful meta-analysis by Davies SJ et al. [7], general cognitive functioning and memory were notably impaired in pregnant women compared with control women, particularly during the third trimester [7]. This early onset of cognitive decline underscores the importance of closely monitoring cognitive health in pregnant women with T1DM throughout their entire pregnancy. The cognitive impairments observed in individuals with T1DM are believed to be associated with alterations in brain glucose metabolism and microangiopathy. These vascular changes can lead to reduced blood flow and oxygen delivery to the brain, resulting in neural damage and impaired cognitive function. Studies have suggested that both hypoglycemia and hyperglycemia contribute to cognitive dysfunction [8,9,10]. According to Cameron F.J. et al. [11], chronic hyperglycemia has a more detrimental effect on the development of cognitive dysfunction than hypoglycemia. Patients with complications such as proliferative retinopathy, nephropathy, neuropathy, hypertension, and cardiovascular autonomic neuropathy often show reduced cognitive functions [12,13,14,15]. Previous research suggests that cognitive impairment is a complication of type 1 diabetes, which is associated with impaired synaptic plasticity and neuron loss due to weakened insulin action [16]. In addition to these diabetes-specific factors, previous studies have emphasized the influence of metabolic variables such as body mass index (BMI) and hormone levels on cognitive function in the general population. As body weight increases, insulin sensitivity decreases, which can lead to metabolic disorders [17]. Leptin plays a crucial role in regulating food intake and energy balance in adults, serving as a satiety factor in the adult brain [18]. Elevated circulating leptin levels have been linked to human cognitive impairment [18]. This may be due to leptin resistance, where the brain becomes less responsive to the hormone’s signals, leading to disruptions in energy balance, metabolic regulation, and cognitive function. Considering that BMI and leptin levels can fluctuate significantly during pregnancy, it is reasonable to suggest that these factors may contribute to cognitive changes observed in women with T1DM during this critical period [19,20]. We hypothesize that pregnancy and the postpartum period may be linked to reduced cognitive function in women with type 1 diabetes mellitus. This study aims to compare the cognitive functions of women with T1DM during and after pregnancy, as well as one year post-delivery. Additionally, we aim to investigate the impact of leptin and body mass index on cognitive function.

## 2. Materials and Methods

### 2.1. Ethical Statements

This study was approved by the Ethics Committee School of Medicine, University of Zagreb. Approval code: No. 380-59-10106-19-111/26 within the scientific project PRE-HYPO No. IP-2018-01-1284. All women in this study provided informed consent.

### 2.2. Study Participants

#### 2.2.1. Inclusion Criteria

In a prospective longitudinal observational cohort study, 84 women diagnosed with type 1 diabetes mellitus before completing 10 gestational weeks were included in this research, conducted from 1 February 2019 to 31 December 2021. At 12 weeks of gestation, 64 women completed an essential cognitive function test (CogniFit). Between three and five days after delivery, 64 mothers completed a cognitive function test. One year after birth, 52 mothers completed cognitive function tests. Pregnant women were provided with intensified therapy using fast-acting insulin (aspart) and long-acting insulin (detemir), ensuring the best care for both mother and child.

The mothers completed the CogniFit test in a quiet room. Cognitive function was assessed in different domains, such as reasoning, memory, attention, coordination, and perception (https://www.cognifit.com, accessed on 15 May 2019). A score between 0 and 200 represents cognitive weakness. A score between 200 and 400 is a low score, although within the average; a score between 400 and 600 is a high score; and a score between 600 and 800 is above the norm.

#### 2.2.2. Exclusion Criteria

To ensure accurate results, pregnant women with T1DM and coexisting proliferative retinopathy, nephropathy, and chronic hypertension were excluded from this study.

### 2.3. Data Collection

#### 2.3.1. Cognitive Test

CogniFit, a cognitive assessment battery, evaluates cognitive function in various areas. The test encompasses reasoning, memory, attention, coordination, and perception. This program comprehensively assesses cognitive well-being, categorizing it as high, moderate, or low, and helps identify strengths and weaknesses in memory, attention, executive functions, planning, and coordination. Reasoning includes subdomains such as planning, processing speed, and shifting. Memory covers subdomains such as phonological short-term memory, contextual memory, naming, short-term memory, non-verbal memory, visual short-term memory, and working memory. Attention involves divided attention, focused attention, inhibition, and updating. Coordination includes visual–motor coordination and reaction speed. Perception encompasses auditory perception, estimation, recognition, spatial perception, visual perception, and visual scanning. In general, cognitive functioning is evaluated across reasoning, memory, attention, coordination, and perception.

#### 2.3.2. Blood Sample Analyses

Maternal vein sera were analyzed for fasting leptin, and HbA1c percentage was measured in maternal blood only.

Glucose levels were quantified using hexokinase on a Cobas C301 analyzer with Roche reagents. The HbA1c levels in whole blood were measured via turbidimetric inhibition immunoassays on a Cobas C501 instrument (Roche, Basel, Switzerland). Leptin serum concentration was determined using a sandwich kit from Tecan, IBL. International (Cat. No. MD53001).

The following parameters were recorded: maternal height (cm) and weight (kg) before pregnancy and pre-pregnancy body mass index (kg/m^2^; BMI) calculated from the pre-pregnancy values.

### 2.4. Statistical Analyses

Statistical analyses were conducted using SPSS version 26 (IBM, Armonk, NY, USA). Categorical data are presented as absolute and relative frequencies. The Kolmogorov–Smirnov test assessed the normality of distribution. Numerical data with normal distributions are described using the mean and standard deviation, while non-normally distributed data are described using the median and interquartile range. Group differences between the normally distributed continuous variables were tested using Student’s *t*-test, while the differences between non-normally distributed continuous variables were tested using the Mann–Whitney U test. Multiple linear regressions examined the association between cognitive function and BMI or leptin concentrations.

Leptin levels were categorized into three groups: 1st group, <14.0 ng/L (<25th percentile); 2nd group, 14.1–39.0 ng/L; and 3rd group, >39.1 ng/L (>75th percentile). Kendall’s coefficient of concordance, a non-parametric measure for ordinal association, evaluated the strength of the relationship between two ordinal variables. The Kruskal–Wallis non-parametric test was used to three groups for a continuous variable. The Bonferroni correction was applied for multiple comparisons.

All *p*-values were two-sided, set at *p* < 0.05.

## 3. Results

Table 1 shows the demographic data of the pregnant women. The average age of the participants was 30.9 years, with a mean diabetes duration of 14.9 years. Most of the pregnant women (60.9%) were over 30 years old, with disease onset generally occurring after ten years of age (60.9%) and having a longer duration than eight years (70.3%). There were 28 (43.8%) overweight and 17 (26.6%) obese pregnant women. In total, 53.8% of them had a high school diploma.

Table 2 displays the cognitive function test results for the pregnant women, mothers after giving birth, and a year after childbirth. A total of 64 patients were tested during pregnancy and after giving birth, while 52 were tested one year after childbirth. Table 3 shows the minimum and maximum score values for each cognitive domain. Upon comparing the score values for each domain during pregnancy, after childbirth, and one year after childbirth, a significant difference was observed in reasoning and attention. Additionally, a noticeable decline in reasoning and attention was found when comparing pregnant women and mothers one year after giving birth. A deterioration in cognitive function for the reasoning and attention domains was observed when comparing mothers after childbirth with those one year postpartum.

Table 3 presents the cognitive function test results for the pregnant women. During the first trimester, above-average scores (600–800) were achieved in the following areas: reasoning by 5 women (5.4%), memory by 1 woman (1.6%), attention by 25 women (40.6%), perception by 7 women (10.8%), and overall cognitive function by 3 women (4.7%). Most of the pregnant women scored in a high range (400–600) across all cognitive function areas. A below-average score was observed in 20 women (31.3%) for memory and 1 woman (1.6%) for coordination. Additionally, 26 women (40.6%) scored in the average range (200–400) for memory and 35 women (54.7%) for coordination.

After childbirth, 64 mothers took the cognitive function test. Above-average scores (600–800) were observed in memory for 3 women (4.7%), attention for 23 women (35.9%), reasoning for 7 women (10.9%), and coordination for 3 women (4.7%). The percentage of mothers with below-average scores slightly decreased to 21 (31.3%) compared with the first trimester (32.8%).

Upon comparing the score values for each domain during pregnancy, after childbirth, and one year after childbirth, a significant difference was observed in reasoning and attention.

One year after childbirth, 52 mothers took the cognitive function test. The number of women with above-average scores decreased to 27 (44.2%), while the number of women with below-average scores significantly increased to 23 (45.3%). Furthermore, the average high and low scores showed slight differences between the first trimester, after childbirth, and one year after childbirth. A deterioration in cognitive function for the reasoning and attention domains was observed when comparing mothers after childbirth to those one year postpartum.

The results of the multivariate linear regression analysis are presented in Table 4. Following adjustments for age and T1DM duration, higher BMI and leptin levels were associated with poorer reasoning, memory, and overall cognitive function in pregnant women with T1DM during the first trimester. These findings suggest that greater BMI and leptin levels in T1DM pregnant women negatively impact reasoning, memory, and overall cognitive function during pregnancy.

Obese pregnant women had significantly lower scores for reasoning than the overweight group (Figure 1).

Pregnant women from the >39.1 ng/L leptin group had lower memory scores than those in the 14.1 ng/L group (Figure 2).

Pregnant women with a normal body weight had significantly better memory than obese women (Figure 3).

Figure 4 shows that pregnant women with leptin values higher than 39.1 ng/L tended to have poor general cognitive function.

Significant differences in total cognitive function scores were found between the >39.1 and 14.1–39 leptin groups. The Kruskal–Wallis test was adjusted with the Bonferroni correction for multiple tests (*p* = 0.015).

Obese individuals with T1DM exhibited the worst cognitive function (Figure 5).

## 4. Discussion

This study is the first to assess cognitive function in pregnant women, postpartum women, and mothers with type 1 diabetes one year postpartum. We found that mothers had lower scores in reasoning and attention one year after giving birth than pregnant women. Earlier research has shown that individuals with T1DM have worse cognitive function than their healthy peers [4,5,6]. Based on these findings, cognitive dysfunction can be considered an important comorbidity of diabetes [21].

Research on cognitive function in non-diabetic pregnant women suggests that cognitive decline during pregnancy is often caused by increased hormone levels [22,23]. Some experts believe that increased estrogen, progesterone, cortisol, thyroxine, prolactin, and human placental lactogen levels during the third trimester have a substantial impact, resulting in reduced cognitive function compared with non-pregnant women [23]. In the available literature, there is no single opinion on the influence of pregnancy hormones on cognitive function. While some believe that pregnancy hormones reduce cognitive function [24,25], others argue that cognitive function either improves or remains unchanged [25].

In this study, pregnant women took a cognitive function test during the first trimester, when hormonal changes had not yet significantly impacted them. The cognitive test conducted after childbirth was strongly influenced by hormonal changes during pregnancy and their rapid decline postpartum. Comparing mothers immediately postpartum with those one year postpartum, we observed worse reasoning and attention scores in the latter group. Other studies have shown that pregnant women, particularly in the third trimester, experience a decline in cognitive function compared with non-pregnant women.

The complexity of cognitive dysfunction in pregnant women with T1DM is influenced not only by pregnancy and childbirth but also by factors related to diabetes. The severity of cognitive impairment in T1DM patients is influenced by the patient’s age and the onset and duration of diabetes [4,6]. In this study, no increased risk was found for low cognitive function in relation to age or diabetes duration. However, early-onset disease (before age 10) and obesity significantly worsen cognitive function compared with later disease onset. Non-pregnant patients with type 1 diabetes have lower cognitive functions than controls without diabetes. A meta-analysis revealed that children with T1DM have poorer cognitive function results than their healthy peers [26]. Adults with T1DM showed poorer results across all cognitive function tests, with cognitive decline more severe in adults than children with T1DM. Early-onset diabetes is associated with lower cognitive test results [27].

Obesity was a significant risk factor for impaired cognitive function. Among 64 pregnant women with T1DM, 8 (12.5%) were obese (BMI ≥ 30 kg/m^2^). Obese pregnant women with T1DM onset before age 10 had significantly lower cognitive function scores. Obese pregnant women were at higher risk for reduced overall cognitive function, memory, and coordination [28,29].

Leptin is a hormone secreted by adipose tissue, which suppresses appetite and regulates energy expenditure [30]. Higher leptin levels were observed in pregnant women with higher BMIs, and these women had significantly higher leptin concentrations in the blood, which were associated with lower cognitive function and memory scores. Chronic central leptin overexpression induces leptin resistance, mimicking many characteristics associated with diet-induced or adult-onset obesity, such as reduced leptin receptors, diminished signaling, and impaired responsiveness to exogenous leptin [31]. Leptin resistance leads to dysfunction in central leptin signaling and hypothalamic impairment, resulting in hippocampal and cortical dysfunction, ultimately leading to impaired cognitive function. In this study, there was a negative correlation between leptin levels and cognitive function scores.

## 5. Conclusions

This study highlighted the importance of monitoring cognitive function in pregnant women with T1DM, particularly those with higher BMIs and leptin levels. Interventions that manage BMI and leptin concentrations could mitigate cognitive decline during and after pregnancy in this population.

## Figures and Tables

**Figure 1 nutrients-16-02751-f001:**
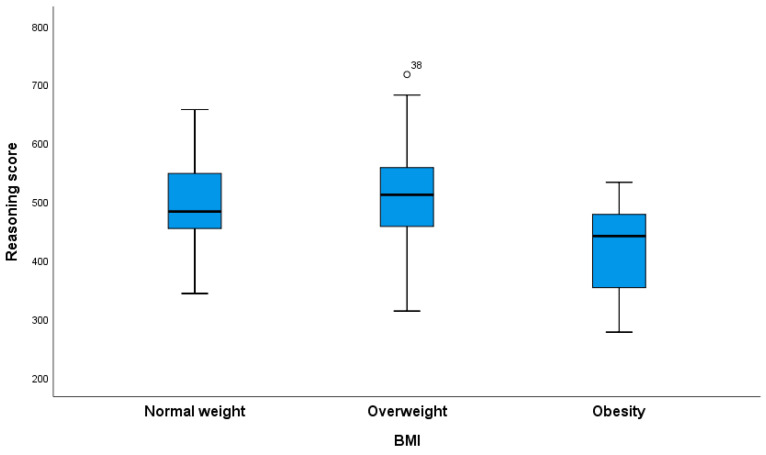
The reasoning scores of pregnant women based on their leptin groups. Significant differences in reasoning scores were found between the overweight and obesity groups (Kruskal–Wallis test with Bonferroni correction for multiple comparisons, *p* = 0.017).

**Figure 2 nutrients-16-02751-f002:**
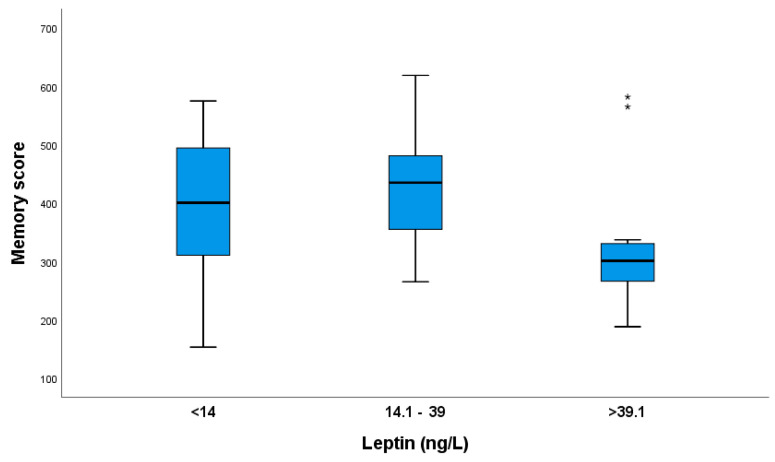
The memory scores of pregnant women based on their leptin groups. Significant differences in memory scores were found between the >39.1 and 14.1–39 ng/L leptin groups. Kruskal–Wallis test with Bonferroni correction for multiple tests, *p* = 0.022. (** Indicates two participants with high leptin levels and high memory scores).

**Figure 3 nutrients-16-02751-f003:**
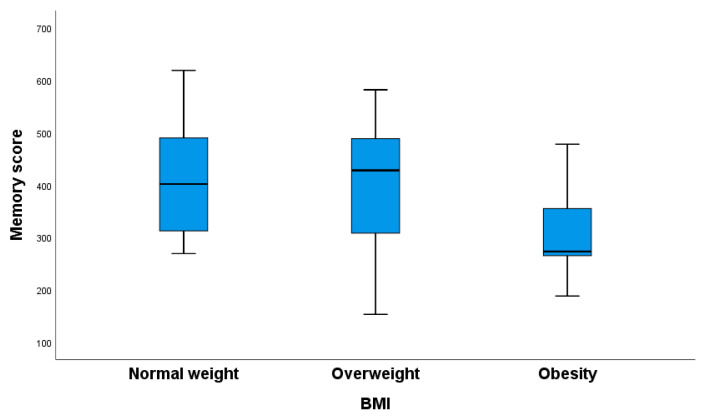
The memory scores of pregnant women based on their body mass index (BMI) groups. Significant differences were found in memory scores between the overweight and obesity groups (*p* = 0.012) and the normal-weight and obesity groups (*p* = 0.032). Kruskal–Wallis test with Bonferroni correction for multiple comparisons.

**Figure 4 nutrients-16-02751-f004:**
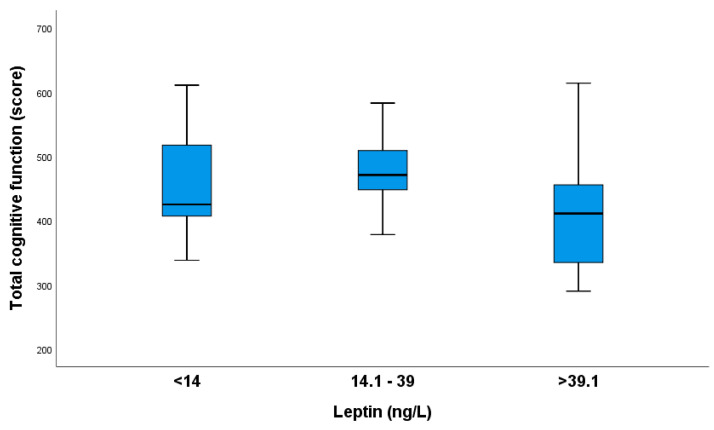
The total cognitive function scores of pregnant women based on their leptin groups.

**Figure 5 nutrients-16-02751-f005:**
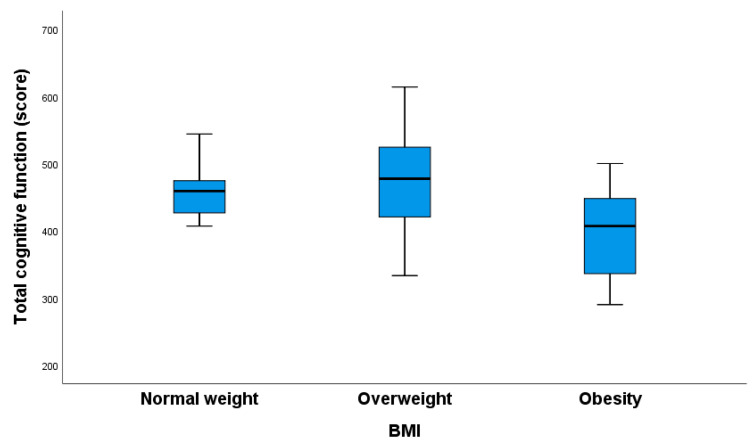
The total cognitive function scores of pregnant women were analyzed based on their leptin groups. Significant differences were found between the obesity and normal-weight groups (*p* = 0.023) and between the overweight and obesity groups (*p* = 0.008). The Kruskal–Wallis test was used, with Bonferroni correction applied for multiple comparisons.

**Table 1 nutrients-16-02751-t001:** Patient demographic data in the first trimester.

Variable	n	Minimum	Maximum	Mean ± SD
Age (years)	64	19	40	30.9 ± 5.2
<30 years	25	19	29	25.6 ± 3.0
≥30 years	39	30	38	34.3 ± 2.9
Duration of T1DM (y)	64	2	37	14.9 ± 9.2
≤8 years	19	2	8	3.5 ± 2.0
>8 years	45	9	37	19.8 ± 6.3
Age of T1DM onset (y)	64	2	36	16.0 ± 9.3
≤10 years	25	2	10	7.3 ± 2.8
>10 years	39	11	36	21.6 ± 7.5
Height (cm)	64	156	183	166.3 ± 5.7
Gestational weight gain (kg)	64	0	22	13.0 ± 4.4
BMI (kg/m^2^) before pregnancy	64	18.1	37.4	23.6 ± 4.7
≤24.9 (kg/m^2^)	44	18.1	24.6	21.4 ± 1.6
25–29.9 (kg/m^2^)	12	25	29.9	27.1 ± 3.7
≥30.0 (kg/m^2^)	8	31	37.4	32.3 ± 1.3
BMI (kg/m^2^) at the time of delivery	64	18.9	43.0	28.0 ± 4.7
≤24.9 (kg/m^2^)	18	18.9	24.8	23.1 ± 1.7
25–29.9 (kg/m^2^)	28	25.3	29.9	27.6 ± 1.6
≥30.0 (kg/m^2^)	17	30	43	34.1 ± 3.2
Education	64			
High school and university	30			
Graduate school	34			
HbA1c (%) 1st trimester	64	5.4	8.0	7.2 ± 0.7
HbA1c (%) 2nd trimester	64	4.6	11.6	6.2 ± 1.1
HbA1c (%) 3rd trimester	64	4.2	7.8	5.9 ± 0.8
Leptin 1st trimester (ng/L)	60	3.7	234.9	22.8 (14.9–39.1)

BMI, body mass index; HbA1c, glycated hemoglobin.

**Table 2 nutrients-16-02751-t002:** Cognitive test results (continuous data).

Cognitive Function in Pregnancy				
Domain	N	Minimum	Maximum	Mean ± SD
Reasoning ^a^	64	277	717	479.8 ± 96.1
Memory	64	153	619	385.6 ± 115.9
Attention ^c^	64	290	745	567.4 ± 102.2
Coordination	64	10	584	285.0 ± 128.0
Perception	64	249	672	480.6 ± 91.8
Total cognitive function	64	228	614	446.4 ± 82.1
Cognitive function postpartum				
Reasoning ^b^	64	110	698	451.8 ± 134.7
Memory	64	159	672	421.0 ± 110.3
Attention ^d^	64	124	738	538.6 ± 128.9
Coordination	64	31	626	299.4 ± 142.3
Perception	64	153	663	485.4 ± 101.7
Total cognitive function	64	157	645	454.4 ± 95.7
Cognitive function one year after delivery				
Reasoning ^a,b^	52	107	674	437.5 ± 120.7
Memory	52	115	710	437.4 ± 128.3
Attention ^c,d^	52	125	736	482.0 ± 141.4
Coordination	52	104	653	299.2 ± 134.8
Perception	52	186	645	450.8 ± 114.0
Total cognitive function	52	159	611	432.7 ± 99.1

Reasoning ^a^, *p* = 0.018; reasoning ^a,b^, *p* = 0.021; attention ^c,d^, *p* = 0.001; Kendall’s coefficient of concordance with Bonferroni correction for multiple tests.

**Table 3 nutrients-16-02751-t003:** Cognitive function test results for the first trimester, postpartum, and one year after delivery (categorical data).

The First Trimester, n = 64				
Domain/Score	600–800 n (%)	400–600 n (%)	200–400 n (%)	0–200 n (%)
Reasoning ^a^	5 (5.4)	46 (71.9)	13 (20.3)	
Memory	1 (1.6)	30 (46.9)	32 (50.0)	1 (1.6%)
Attention ^c^	25 (40.6)	34 (53.1)	4 (6.3)	
Coordination		13 (20.3)	31 (48.4)	20 (31.3)
Perception	7 (10.8)	43 (67.2)	14 (21.9)	
Total cognitive function	3 (4.7)	46 (71.9)	15 (23.4)	
Postpartum (after delivery), n = 64				
Reasoning ^b^	7 (10.9)	40 (62.5)	14 (21.9)	3 (4.7)
Memory	3 (4.7)	34 (53.1)	26 (40.6)	1 (1.6)
Attention ^c,d^	23 (35.9)	33 (51.6)	7 (10.9)	1 (1.6)
Coordination	3 (4.7)	12 (18.8)	35 (54.7)	14 (21.9)
Perception	9 (14.1)	45 (70.3)	9 (14.1)	1 (1.6)
Total cognitive function	4 (6.3%)	41 (64.1%)	18 (28.1%)	1 (1.6%)
One year after delivery, n = 52				
Reasoning ^a,b^	1 (1.9)	36 (69.2)	13 (25.0)	2 (3.8)
Memory	7 (13.5)	28 (53.8)	15 (28.8)	2 (3.1)
Attention ^c,d^	13 (25.0)	26 (50.0)	11 (21.2)	2 (3.8)
Coordination	2 (3.8)	11 (21.2)	24 (45.2)	15 (28.8)
Perception	4 (7.7)	31 (59.6)	16 (30.8)	1 (1.9)
Total cognitive function	1 (1.9)	37 (71.2)	13 (25.0)	1 (1.9)

Reasoning ^a^, *p* = 0.021; reasoning ^a,b^, *p* = 0.018; attention ^c^, *p* = 0.032; attention ^c,d^, *p* = 0.001. Kendall’s coefficient of concordance with Bonferroni correction for multiple tests.

**Table 4 nutrients-16-02751-t004:** Linear regression model associating serum leptin and BMI measures with cognitive function.

Model		β (95% CI)	*p*
Reasoning	Leptin	−0.309 (−0.146; −0.015)	0.017
Reasoning	BMI	−0.251 (−0.044; −0.024)	0.044
Memory	Leptin	−0.386 (−0.146; −0.031)	0.002
Memory	BMI	−0.244 (−0.020; −0.000)	0.046
Attention	Leptin	−0.292 (−0.137; −0.009)	0.027
Attention	BMI	−0250 (−0.052; −0.001)	0.052
Coordination	Leptin	−0.214 (−0.092; −0.009)	0.109
Coordination	BMI	−0.173 (−0.015; −0.003)	0.169
Perception	Leptin	−0.203 (−0.121; 0.016)	0.127
Perception	BMI	−0.059 (−0.016; 0.010)	0.641
Total CogniFit	Leptin	−0.382 (−0.190; −0.040)	0.003
Total CogniFit	BMI	−0.218 (−0.027; −0.001)	0.078

Adjusted with age and T1DM duration. BMI, body mass index.

## Data Availability

The data presented in this study are available upon request from the corresponding author, Josip Delmis.

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
