# Peer review of "Cognitive Function during and after Pregnancy and One-Year Postpartum in Type 1 Diabetes: A Longitudinal Study"

_nutrients, 2024, doi:10.3390/nu16162751_

Round 1

Reviewer 1 Report

Comments and Suggestions for Authors

The paper “Cognitive Function during and after Pregnancy, and one Year Postpartum in Type 1 Diabetes: A Longitudinal Study” investigates cognitive function in women with type 1 diabetes (T1DM) during pregnancy, immediately after delivery, and one year postpartum, examining the influence of leptin and body mass index (BMI). The research is interesting and meaningful, but there are some questions that need to be further improved or explained.

Comments:

Q1. The abstract section does not necessarily require direct expressions of those words including “methods”, “results”, “conclusions”.

Q2. The abstract fails to accurately represent the research background and research significance.

Q3. This paper mainly studies the cognitive status of women during pregnancy, so what is the purpose of introducing the cognitive status of children born to patients with high sugar? The introduction still does not clearly present the research background, research content and research significance of this paper, which need to be further improved.

Q4. The main functions of leptin include suppressing appetite, increasing energy consumption, reducing fat synthesis and storage, and reducing body weight. It is one of the important substances in the body's resistance to obesity. Higher BMI and leptin levels appeared in the results of this paper concurrently. The presence of any conflicting findings should be accompanied by a well-founded justification in the appropriate section.

Q5. The discussion part is not in-depth enough to reflect the author's effective combination of the existing literatures and the results of this paper, which needs to be further improved. In addition, is cognitive impairment caused by a single factor of obesity, pregnancy, diabetes or a combination of these factors? Are higher levels of leptin less likely to cause symptoms of obesity? What is the correlation between diabetes and leptin? All these points need in-depth discussion and analysis, and the references in this paper are relatively insufficient, especially that in the recent three years.

Q6. The research in this paper has certain innovation and research significance, but the authors fails to give clear expressions, so it needs to be revised very seriously.

Author Response

The paper “Cognitive Function during and after Pregnancy, and one Year Postpartum in Type 1 Diabetes: A Longitudinal Study” investigates cognitive function in women with type 1 diabetes (T1DM) during pregnancy, immediately after delivery, and one year postpartum, examining the influence of leptin and body mass index (BMI). The research is interesting and meaningful, but there are some questions that need to be further improved or explained.

Q1. The abstract section does not necessarily require direct expressions of those words including “methods,” “results,” “conclusions.”

Authors' Response: Thank you for your suggestion. We have revised the abstract to better articulate the research background and significance.

Q2. The abstract fails to accurately represent the research background and research significance.

Authors' Response: Specifically, we have highlighted that this study aims to compare the cognitive function of women with T1DM during and after pregnancy, as well as one year postpartum, and to investigate the impact of leptin and body mass index on cognitive function.

Q3. This paper mainly studies the cognitive status of women during pregnancy, so what is the purpose of introducing the cognitive status of children born to patients with high sugar? The introduction still does not clearly present the research background, research content, and research significance of this paper, which need to be further improved.

Authors' Response: Thank you for pointing this out. We have removed the content related to the cognitive status of children born to patients with high blood sugar to maintain focus on the primary objective of the study. We have also revised the introduction to clearly present the research background, content, and significance, aligning it more closely with the study's aims.

Q4. The main functions of leptin include suppressing appetite, increasing energy consumption, reducing fat synthesis and storage, and reducing body weight. It is one of the important substances in the body's resistance to obesity. Higher BMI and leptin levels appeared in the results of this paper concurrently. The presence of any conflicting findings should be accompanied by a well-founded justification in the appropriate section.

Authors' Response: Thank you for your insightful comments. We have improved the introduction to address the role of leptin and BMI in cognitive function. The revised section now discusses how elevated leptin levels, often associated with leptin resistance, may contribute to cognitive impairment. We have also included relevant literature to provide a well-founded justification for our findings, highlighting the complex interplay between leptin, BMI, and cognitive function in both the general population and women with T1DM.

Q5. The discussion part is not in-depth enough to reflect the author's effective combination of the existing literature and the results of this paper, which needs to be further improved. In addition, is cognitive impairment caused by a single factor of obesity, pregnancy, diabetes, or a combination of these factors? Are higher levels of leptin less likely to cause symptoms of obesity? What is the correlation between diabetes and leptin? All these points need in-depth discussion and analysis, and the references in this paper are relatively insufficient, especially in the recent three years.

Authors' Response: We appreciate your detailed feedback on the discussion section. We have significantly expanded the discussion to more deeply analyze the potential interactions between obesity, pregnancy, diabetes, and cognitive function. We have also explored the role of leptin in these processes and discussed its correlation with diabetes. Furthermore, we have updated the references to include more recent studies from the past three years, ensuring that our discussion is both current and comprehensive.

Q6. The research in this paper has certain innovation and research significance, but the authors fail to give clear expressions, so it needs to be revised very seriously.

Authors' Response: Thank you for your constructive critique. We have undertaken a thorough revision of the manuscript to clearly express the innovative aspects and significance of our research. We are committed to presenting our findings in a clear and impactful manner.

We hope these revisions address your concerns and improve the clarity and depth of our manuscript. We greatly appreciate your thoughtful comments and suggestions, which have undoubtedly strengthened our work.

Reviewer 2 Report

Comments and Suggestions for Authors

The article is interesting and addresses an important topic by researching cognitive function during and after pregnancy in women with type 1 diabetes. The study shows that cognitive functions decline one year postpartum and that obesity and high leptin levels can further exacerbate these issues. The paper suggests managing BMI and leptin levels to mitigate cognitive decline. The article is engaging, the references are relevant, the plagiarism level is low, and I support its publication.

However, I have some questions: Is the sample size of 64 sufficient to draw general conclusions? How does the study plan to examine the long-term consequences, and what other important factors were considered in the analysis of the impact on cognitive function (did the study adequately account for hormonal changes during pregnancy)?

There are a couple of minor typographical errors, such as "Glucose levels werequantified," and it would be helpful to include an abbreviation list under the tables (e.g., T1DM, BMI, HbA1c).

Comments on the Quality of English Language

-

Author Response

R: The article is interesting and addresses an important topic by researching cognitive function during and after pregnancy in women with type 1 diabetes. The study shows that cognitive functions decline one year postpartum and that obesity and high leptin levels can further exacerbate these issues. The paper suggests managing BMI and leptin levels to mitigate cognitive decline. The article is engaging, the references are relevant, the plagiarism level is low, and I support its publication.

Authors' Response: We sincerely thank the reviewer for the kind words and support. We are pleased that you find our study both interesting and relevant.

R: However, I have some questions: Is the sample size of 64 sufficient to draw general conclusions?

Authors' Response: We appreciate your question regarding the sample size. We used G*Power 3.1.9.7 to perform a power analysis to ensure that our sample size was adequate for detecting significant differences. Specifically, for the Wilcoxon-Mann-Whitney test (comparing two groups) regarding cognitive function between pregnancy and one year postpartum, assuming a normal distribution, an alpha error probability of 0.05, and a power (1-β) of 0.85, a total sample size of 12 patients was required. For analyzing attention between pregnancy and postpartum under similar conditions, the required sample size was 41 patients. Given these calculations, we believe that our sample size of 64 participants is sufficient to draw reliable conclusions.

R: How does the study plan to examine the long-term consequences, and what other important factors were considered in the analysis of the impact on cognitive function (did the study adequately account for hormonal changes during pregnancy)?

Authors' Response: Thank you for this important question. Our research is ongoing, and we plan to continue monitoring cognitive function in relation to various factors, including age, duration of diabetes, continuous glucose monitoring (CGM) data, the onset of diabetes, BMI, and leptin levels.

Regarding the influence of pregnancy hormones on cognitive function, the current literature does not offer a consensus. Some studies suggest that pregnancy hormones, such as estrone, estradiol, estriol, progesterone, and 17-hydroxyprogesterone, may reduce cognitive function. Conversely, other studies suggest an improvement or no significant change in cognitive function during pregnancy and postpartum. In our study, we monitored chorionic gonadotropin levels up to 16 weeks of pregnancy and did not find a significant connection between these hormonal changes and cognitive function.

We hope this clarifies our approach and the considerations taken into account during the study.

We are grateful for your thoughtful questions and feedback, which have allowed us to provide further clarity and detail in our responses.

Round 2

Reviewer 1 Report

Comments and Suggestions for Authors

I have no additional comments for the manuscript (nutrients-3166539). 

The manuscript could be accepted if the content and format of the abstract and references sections meet the journal standards.